# Surgical Treatment Using Sinus Tarsi Approach with Anterolateral Fragment Open-Door Technique in Sanders Type 3 and 4 Displaced Intraarticular Calcaneal Fracture

**DOI:** 10.3390/ijerph181910400

**Published:** 2021-10-02

**Authors:** Jaeho Cho, Jahyung Kim, Eun Myeong Kang, Jeong Seok Lee, Tae-Hong Min, Sung Hun Won, Young Yi, Dong-il Chun

**Affiliations:** 1Department of Orthopaedic Surgery, Chuncheon Sacred Heart Hospital, Hallym University, 77, Sakju-ro, Gyo-dong, Chuncheon 24262, Korea; hohotoy@nate.com; 2Department of Orthopaedic Surgery, Soonchunhyang University Seoul Hospital, 59 Daesagwan-ro, Yongsan-gu, Seoul 04401, Korea; hpsyndrome@naver.com (J.K.); 129741@schmc.ac.kr (E.M.K.); 124856@schmc.ac.kr (J.S.L.); minth916@gmail.com (T.-H.M.); orthowon@gmail.com (S.H.W.); 3Department of Orthopaedic Surgery, Seoul Foot and Ankle Center, Inje University Seoul Paik Hospital, 85, 2-ga, Jeo-dong, Jung-gu, Seoul 04551, Korea; 20vvin@naver.com

**Keywords:** calcaneal fracture, displaced intraarticular fracture, sinus tarsi approach, extensile lateral approach

## Abstract

Although various outcomes of the sinus tarsi approach have been reported, these are limited to the Sanders type 2 displaced intraarticular calcaneal fractures (DIACF) because of the limited visibility of the posterior facet joint. In this study we aimed to (1) introduce a sinus tarsi approach combined with an anterolateral fragment open-door technique that enables adequate visibility of the innermost and middle portion of the posterior facet joint, and (2) evaluate the radiographic and clinical outcomes of the patients treated with that technique. This is a retrospective case-series study performed on medical records of 25 patients who presented with the Sanders type 3 or 4 DIACF and were treated with the sinus tarsi approach. The radiologic measurements showed significant corrections of the Bohler’s angle, calcaneal width, length, height, and articular step-off in both X-rays and CTs in the last follow-up period. The mean AOFAS score was 90.08 ± 6.44 at the last follow-up. Among all the follow-up patients, two cases (8%) had acute superficial infections, and no other wound complications occurred. Therefore, we suggest that the Sanders type 3 or 4 DIACF could be successfully treated with the proposed technique with low complications and bring out effective clinical and radiologic outcomes.

## 1. Introduction

Controversies exist regarding an adequate surgical approach to produce optimal postoperative outcomes for a calcaneus fracture [1]. For decades, an extensile lateral approach has been considered to be a gold-standard modality to access a calcaneus fracture, providing both complete exposure of the calcaneus and convenience of plate application [2]. However, such an invasive approach has a critical drawback in terms of wound complications [3,4]. In addition, hematoma formation, sural nerve neuropathy, and complex regional pain syndrome have also been reported [5,6,7]. In an effort to minimize the inevitable complications of the extensile lateral approach, a variety of alternative, minimally invasive approaches have been developed [8,9,10].

Among these less-invasive approaches, the sinus tarsi approach is known to be the most commonly used in the literature, having been modified into various forms since it was first introduced by Palmer [11,12]. Competitive clinical and radiologic outcomes of the sinus tarsi approach compared with the extensile lateral approach have been reported in previous studies, simultaneously minimizing the wound complication rates in the tongue-type or less severely displaced fractures, that is, mainly the Sanders type 2 fracture [13]. In addition, there were reports that the operation time was relatively short and the functional outcome was excellent compared to the extensile lateral approach [8,14]. However, many surgeons are reluctant to use a sinus tarsi approach in the Sanders type 3 or 4 fractures [15] because of its limited visibility of the innermost or middle portion of the posterior facet fragments risking postoperative wound complications.

In this study we introduce a sinus tarsi approach combined with an anterolateral fragment open-door technique in more severely displaced intra-articular calcaneal fractures (DIACF), that is, a Sanders type 3 or 4 fracture, that enables adequate visibility of the innermost and middle portion of the posterior facet joint followed by its anatomical restoration. Furthermore, with the aid of this technique, we aimed to evaluate the radiographic and clinical outcomes of the Sanders type 3 or 4 DIACF treated with a sinus tarsi approach. In this perspective, we hypothesized that if the blind spots of the posterior facet could be overcome, a gratifying outcome could be achieved, even in the Sanders type 3 or 4 displaced intraarticular calcaneal fractures, through a sinus tarsi approach.

## 2. Materials and Methods

This study was approved by the Institutional Review Board at our institution, and written informed consent for the publication of this report was obtained from all the included patients.

### 2.1. Patients

This is a multicenter retrospective study that collected data from patients at two university hospitals. From January 2016 to March 2019, we included 33 consecutive patients who presented with a Sanders type 3 or 4 DIACF and were treated with a sinus tarsi approach combined with an anterolateral fragment open-door technique. After exclusion of the patients who followed up for less than 1 year, ultimately 25 patients were enrolled through the exclusion criteria.

### 2.2. Surgical Technique

Patients were given general or spinal anesthesia and placed in a lateral decubitus position on the operating-room table. Prior to elevation of the tourniquet, the anatomic structures that would be important to be used as landmarks during the procedure were outlined: the border of the distal fibula and the calcaneocuboid joint (Figure 1). An approximate 7 cm skin incision was made horizontally from the distal tip of the fibula to the level distal to the calcaneocuboid joint. Such a horizontal, anteriorly extended incision makes it feasible to see the wide range of the subtalar joint from the calcaneocuboid joint to the posterior facet of the calcaneus, simultaneously avoiding the risk of iatrogenic sural-nerve injury [16]. A deep dissection was continued until the peroneal tendons were identified, retracted inferiorly, and kept within the tendon sheath.

After the lateral capsule and calcaneofibular ligament (CFL) was incised to open the subtalar joint, we identified the anterolateral wall fragment, including the anterior process of the calcaneus. In order to more easily see the posterior facet, a 4.5 mm Steinmann pin was first inserted horizontally medial to lateral at the posterior aspect of the calcaneus, followed by gentle manual posterior traction to restore the length of the impacted calcaneus. Then, after separating attached soft tissues, the anterolateral wall fragment was evaluated for whether a fracture line was adequate for retraction. In case there was no fracture line on the anterolateral wall, or the fracture fragment was not broad enough to create a window, additional osteotomy was performed using a chisel. Next, the anterolateral wall fragment was retracted anterosuperiorly as a bony flap in an open-door manner, which enhanced observation of the middle and innermost fragments of the posterior facet that became possible (Figure 2).

For reduction of the fracture, first a 4.5 mm Steinmann pin was inserted posteroanteriorly into the calcaneal tuberosity, and gentle inferior traction with varus-valgus leverage was applied manually to restore the height of the calcaneus. Then, a periosteal elevator was inserted through the opened anterolateral wall to the undersurface of the displaced posterior facets. Under direct and entire visualization of the posterior facet, the impacted middle fragments and lateral fragment were aligned with the posteromedial sustentacular fragment. These fragments were then temporarily fixed with two Kirschner wires inserted from the lateral posterior facet fragment toward the sustentaculum fragment. Following the confirmation of the reduction, the posterior facet joint was definitely fixed with 4.0 mm cannulated or cancellous screws (Figure 3).

After again confirming the aligned posterior facet by direct visualization through the opened window and C-arm image intensifier, we realigned the hindfoot varus angulation, if present, under a C-arm image intensifier axial view with the valgus force applied with the previously inserted Steinmann pin. Once the adequate hindfoot alignment was checked, one or two fully threaded 6.5 mm or 7.0 mm cannulated screws were inserted percutaneously from the posterior heel in the anterior direction, simultaneously preventing depression of the elevated and fixed posterior facet fracture. Then, the void space underneath the posterior facet was filled with allogenic chip bone for additional support, followed by closing the opened anterolateral wall using a bone impactor. In the cases with significant lateral-wall bulging, an anatomical plate was additionally inserted and fixed into the incision subcutaneously for supplementary lateral stability of the calcaneus.

### 2.3. Postoperative Management

Dressings were evaluated every day until the wound became stable and changed every two or three days afterwards. If there were no issues with the wound status, sutures were removed two weeks after surgery. A tolerable range of motion exercise was initiated as soon as pain was relieved, and a “circle draw” exercise to restore subtalar joint range of motion was initiated after suture removal. A short leg splint was applied for a duration of four weeks postoperatively regardless of the fixation device. Subsequently, patients were encouraged to bear a tolerable, partial weight on their foot with an ankle brace. Full weight-bearing was allowed if the signs of a bony union were observed on the radiograph around 8 to 10 weeks postoperatively.

Lateral, Broden’s, and axial calcaneal radiographs were obtained in the follow-up period postoperatively, and a CT was taken postoperatively (Figure 4, Figure 5 and Figure 6). In terms of clinical outcomes, patients were asked to answer the questionnaire form including the Visual Analogue Scale (VAS) score and the American Orthopaedic Foot and Ankle Society (AOFAS) ankle/hindfoot score in the last follow-up period. These scores were collected and analyzed retrospectively along with incidences of postoperative complications through medical record reviewing.

### 2.4. Radiologic Measurement

We obtained pre- and postoperative standard X-rays with lateral, Broden’s, and axial views and CTs of the feet (Figure 4 and Figure 5). Bohler’s angle was measured using the highest points of the calcaneal tuberosity, subtalar joint, and anterior process [17]. Gissane’s angle is the angle formed by the posterior facet and the line from the calcaneal sulcus to the tip of the anterior process of the calcaneus [17]. The length of the calcaneus was measured on the lateral view from the most posterior point of the tuberosity to the center of the calcaneocuboid joint [18]. The height of the calcaneus was also measured on the lateral view by a line perpendicular to the calcaneal axis to the highest point of the posterior facet [19]. The width of the calcaneus was measured on the axial view as the length of a perpendicular line connecting two parallel lines drawn tangent to the widest part of the calcaneal tuberosity [17]. The articular step-off of the posterior facet was measured on a Broden’s view radiograph [20]. Furthermore, it was also measured in a CT, which was defined as the maximal intra-articular step-off of the posterior facet on the sagittal or coronal views, as in the Sanders classification [21].

All the measurements were performed in the same period by two independent orthopedics residents, both of whom did not participate in the surgery and were blinded to each other’s findings. Two weeks later, the measurements were repeated in the same fashion in order to determine the intra-observer reliability.

### 2.5. Statistical Analysis

All data were analyzed by R version 3.3.1 (‘Optimal Cutpoints’ packages) and were described as mean ± standard deviation. We performed a normality test through the Shapiro–Wilk Test. The preoperative and 1-year follow-up calcaneal radiologic measurements and the pre- and postoperative CT measurements were compared, respectively, using the paired t test or Wilcox-signed rank test. Statistical significance was set at *p* < 0.05. Inter-observer and intra-observer reliabilities were obtained for all measurements using the intraclass correlation coefficient (ICC). ICCs of 0.81 to 1.00, 0.61 to 0.80, 0.41 to 0.60, 0.21 to 0.40, and 0.00 to 0.20 were interpreted as excellent, good, moderate, fair, and poor, respectively [22].

## 3. Results

The average age was 55.84 ± 11.89 (27–76) years; 22 (88%) were male and 3 (12%) were female. For the mechanism of injury, most of the patients presented with a fall from a height; 4 (16%) of them were low (<1 m), 14 (56%) of them were intermediate (1–2 m), 6 (24%) were high (>2 m) height falls, and 1 patient presented after a traffic accident. Based on the Sanders classification, 18 (72%) feet were type 3 fractures, and 7 (28%) feet were type 4 fractures [23]. The average duration from time of injury to the surgery was 3.28 ± 1.86 days, ranging from 0 to 8 days, and the average follow-up period was 15.28 ± 6.50, ranging from 12 to 40 months. Demographic data of the enrolled patients are described in Table 1.

The inter- and intra-observer ICCs for all the radiographic measurements showed excellent reliability except for the preoperative and last follow-up Gissane’s angles. The lateral, Broden’s, and axial radiographs of the calcaneus showed good reduction and fixation, and significant corrections of the Bohler’s angle, calcaneal width, length, height, and articular step-off in both Broden’s view and the CT from the preoperative to the last follow-up period (*p* < 0.001). However, no significant differences were found in the Gissane’s angle between the postoperative and the last follow-up period (Table 2). Among all the follow-up patients, two cases (8%) had acute superficial infections on the wound, which eventually healed after persistent dressing and administration of antibiotics. No other wound problem or iatrogenic sural-nerve injury occurred. In terms of clinical outcomes, the VAS score was 0.92 ± 0.95 (range 0 to 3) and the AOFAS score was 90.08 ± 6.44 (range 76 to 100) in the last follow-up period (Table 3).

## 4. Discussion

In line with the advances in surgical techniques, operative treatment is gaining in preference among orthopedic surgeons as the treatment of choice for the calcaneus fracture with articular displacement [24]. In terms of surgical treatment, the primary goals of open reduction and internal fixation are considered to be re-establishment of the normal anatomy of the posterior facet, narrowing of the width of the calcaneus to prevent lateral impingement of the peroneal tendons, and re-establishment of the normal height of the calcaneus [8]. Fulfilling such preconditions, the extensile lateral approach has been used as a gold-standard modality because it allows the fragments to be seen and provides subsequent ease of reduction and fixation along with a large plate that allows lateral stability [25]. Nevertheless, such an invasive procedure is exposed to devascularization of the calcaneus, a broad surgical field vulnerable to contamination, dead space produced under the fasciocutaneous flap, and increased operation time because of the complexity of the process [7]. Such drawbacks have pushed surgeons to try to find an alternative, less-invasive approach to the calcaneus that may lead to superior clinical outcomes.

The sinus tarsi approach, considered to be the most commonly used minimally invasive approach, produces results comparable to those of the extensile lateral approach with minimal wound problems [6,26]. It is widely known to be indicated for the less severe DIACFs, that is, most of the Sanders type 2, or only a few of the type 3, because it does not allow a clear sight of the posterior facet.

However, a recent cadaveric study has proven that by excluding the superomedial quadrant, as much of the posterior facet could be exposed by means of the sinus tarsi approach as by the extensile lateral approach [27]. Based on such a finding, we hypothesized that if the limited visibility of the superomedial corner of the posterior facet could be overcome, a gratifying outcome could be achieved, even in the Sanders type 3 or 4 DIACF through a sinus tarsi approach.

After much consideration, we concluded that the main reason for the limited vision would be the protuberant anterolateral fragment involving the anterior process of the calcaneus. Because it produces blind spots in the medial portion of the posterior facet when inspected through the sinus tarsi approach incision, we postulated that a wider field of vision would be secured if the anterolateral fragment were put aside (Figure 7). Therefore, we first extended the skin incision anteriorly, reaching the calcaneocuboid joint in order to expose the anterolateral fragment. In an effort to keep the anteriorly extended incision from interacting with the course of the sural nerve, a skin incision was made horizontally parallel to the sole rather than toward the 4th metatarsal base [16]. After the exposure of the calcaneocuboid joint, we separated the anterolateral fragment from the surrounding soft tissue and retracted it anterosuperiorly as a bony flap, which we called an anterolateral fragment open-door technique (Figure 8). Using this maneuver, we eliminated the blind spots along the innermost posterior facet, leading to satisfactory restoration of the articular congruence in the Sanders type 3 or 4 DIACF. Adequate elevation of the depressed fragments of the posterior articular facet could be assessed directly through the window created by the retracted anterolateral fragment along with use of a C-arm image intensifier. Among the procedures performed before the complete establishment of the sinus approach in our institutions, a dry arthroscope was used to confirm the restoration of the posterior facet articular surface (Figure 9). Different from the usual arthroscopic procedure, mostly used for the ankle, which necessitates saline solution [28,29], an arthroscopic procedure in this case has to be performed in a dry manner for two reasons: first, the presence of a surgical opening, if it is a mini-invasive approach, prevents the creation of an internal pressure; second, an increased peri-calcaneal pressure could create damage to the neighboring anatomical structures in a confined space suffering from a hematoma produced by the fracture.

Apart from anatomical reduction of the articular surface, restoration and maintenance of the normal height of the calcaneus are also considered important requisites of successful surgical outcomes, and several studies have shown that restoration of the Bohler’s angle leads to better functional outcomes [30,31]. In addition, because the space underneath the elevated posterior facet becomes void, depression of the reduced articular surface could occur postoperatively during weight bearing or rehabilitation [21]. In this study, the void trabecular space of the calcaneus could become more apparent because of the retracted anterolateral wall fragment, which might increase the risk of calcaneal height subsidence. In order to overcome such a possibility, allogenic bone grafts were filled inside the hollow space in order to support the reduced posterior facet. Although no consensus exists on the Bohler and Gisanne angles as reliable radiologic parameters to represent the calcaneal height status [32], significant restoration of the Bohler’s angle at the last follow-up would imply the effectiveness of our procedure.

In addition to the satisfactory postoperative outcomes, rather few complications have been detected. Despite its minimally invasive nature, the wound complication rate has been reported to range from 0 to 15.4% using the sinus tarsi approach [33]. In terms of the Sanders type 4 DIACF, Lin et al. [34] reported a case-control study between the sinus tarsi approach and extensile lateral approach; wound healing complications totaled 14.28% in the sinus tarsi approach, although the number was significantly lower compared with 34.04% in the conventional extensile lateral approach. On the contrary, only two (8%) cases of superficial wound complications were detected after the surgery in this study. Although the incision in our procedure was relatively long, it has only been extended anteriorly and ceased posteriorly at the level of the fibular tip, preserving the lateral calcaneal artery, the main known source of vascularization of the lateral fasciocutaneous layer. In addition, compared to the previous studies using a space-occupying calcaneal plate for the bony fixation, sufficient fixation could be accomplished with multiple screws instead in the present study, which might have attributed to minimizing the wound complications. Lastly, surgery was carried out as soon as the skin condition became stable enough to suture with an average of 3.28 days from the fracture. However, it is not clear whether it contributed to a lower complication rate because correlation between timing of surgery from the fracture and wound complication remains unclear, especially in the sinus tarsi approach [35,36]. Regarding the sural nerve injuries, none occurred during the surgery, probably because the horizontal incision avoided the course of the sural nerve. Furthermore, no patients required subtalar arthrodesis up until now. When we put all these results together, we suggest that the sinus tarsi approach combined with the anterolateral fragment open-door technique in this study could be performed satisfactorily with minimal risk of postoperative complications.

In terms of the Sanders type 3 or 4 DIACF, since the talus continues to penetrate through the calcaneal body even after creating a secondary fracture line, the lateral wall becomes separated from the posterior facet in most cases, which is the so-called lateral wall blow-out [19]. In order to adequately manage the lateral wall blow-out, the extensile lateral approach is considered to be advantageous over the sinus tarsi approach by providing wider exposure of the lateral calcaneal wall and enough space to place an anatomical plate, which may reduce the bulged lateral wall [27,37]. However, adequate restoration of the calcaneal width has been accomplished with the sinus tarsi approach in this study. Although not routinely used in this study, an anatomical plate could also be inserted subcutaneously in the presence of a significant lateral wall bulging by means of an anteriorly extended incision. Therefore, we believe that the bulging of the lateral wall could also be reduced by means of the sinus tarsi approach.

The limitation of the present study was that it was a retrospective case-series study of only 25 patients without a control group to compare the results. Therefore, succeeding studies with an increased sample size to compare the sinus tarsi approach with the extensile lateral approach in the Sanders type 3 or 4 DIACF would be needed to verify the result of this study. In addition, a relatively short-term follow-up period with a minimum of 12 months might have precluded the observation of long-term complications and prognosis. Consequently, a long-term follow-up design focused on the complications such as malunion or subtalar joint traumatic arthritis would have to be performed in the future.

## 5. Conclusions

In this study, we introduced the sinus tarsi approach using an anterolateral fragment open-door technique, which enabled adequate visibility of the innermost portion of the posterior facet, leading to satisfactory anatomic reduction of the articular surfaces. Therefore, we suggest that the Sanders type 3 or 4 DIACF could be successfully treated with the use of our proposed technique by minimizing the postoperative complications and bringing out effective clinical and radiological outcomes.

## Figures and Tables

**Figure 1 ijerph-18-10400-f001:**
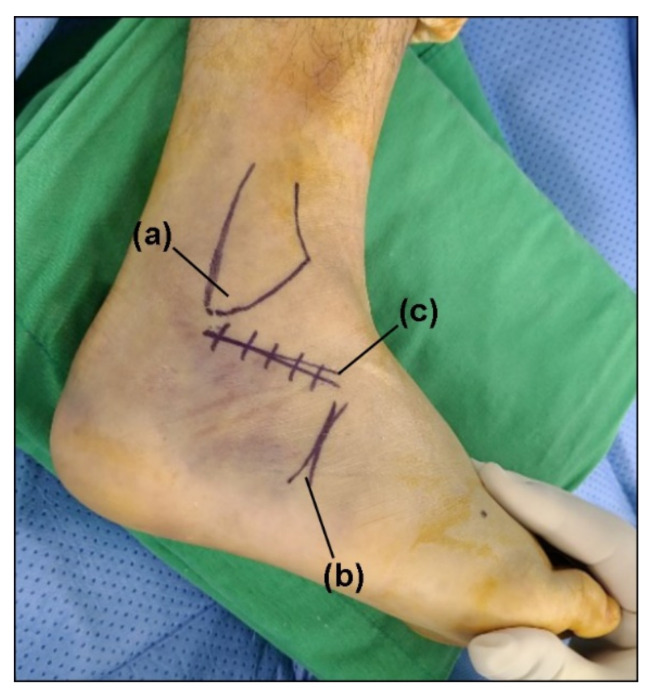
The skin incision; (a) lateral malleolus, (b) calcaneocuboid joint, and (c) incision marker.

**Figure 2 ijerph-18-10400-f002:**
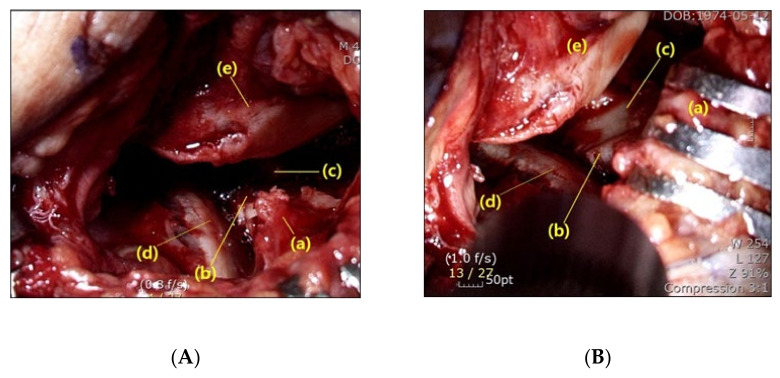
Intraoperative visualization of the subtalar joint. (**A**) Limited visualization of the middle (b) and innermost (c) fragment of the posterior facet due to the presence of the anterolateral fragment (a). (d: lateral fragment of the posterior facet, e: talus) (**B**) Enhanced observation of the middle (b) and innermost (c) posterior facet becomes possible after retraction of the anterolateral fragment (a) (d: lateral fragment of the posterior facet, e: talus).

**Figure 3 ijerph-18-10400-f003:**
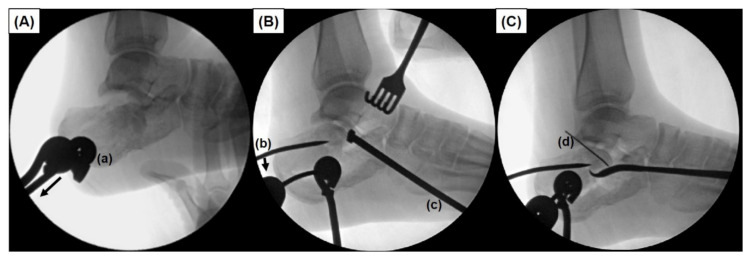
Intraoperative fluoroscopic images describing reduction of a displaced fracture. (**A**) A 4.5 mm Steinmann pin (a) is first inserted horizontally medial to lateral at the posterior aspect of the calcaneus, followed by gentle manual posterior traction (Arrow). (**B**) A 4.5 mm Steinmann pin (a) is inserted posteroanteriorly into the calcaneal tuberosity, and gentle inferior directed leverage force is applied manually by means of a joystick maneuver (b) to restore the height of the calcaneus. A periosteal elevator (c) is inserted through the opened anterolateral wall to the undersurface of the displaced posterior facets. (**C**) Under direct and entire visualization of the posterior facet, reduced fragments are temporarily fixed with a Kirschner wire (d) inserted from the lateral posterior facet fragment toward the sustentaculum fragment.

**Figure 4 ijerph-18-10400-f004:**
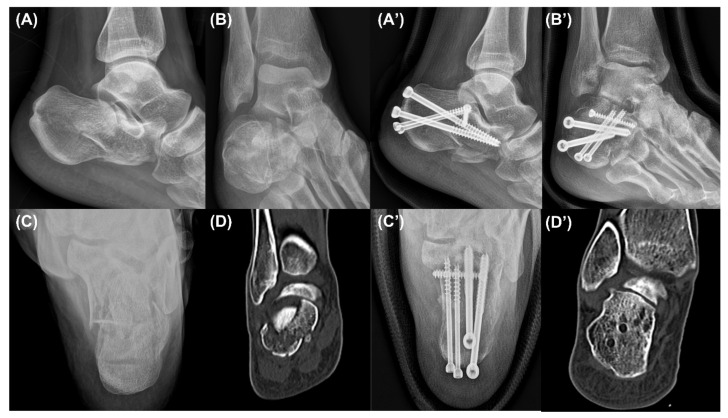
Case 1. Preoperative lateral (**A**), Broden’s (**B**), and axial (**C**) radiographs and a coronal computed tomography image (**D**) showing a Sanders type 4 intra-articular calcaneal fracture. The lateral (**A’**), Broden’s (**B’**), and axial (**C’**) radiographs and a coronal computed tomography image (**D’**) taken in the last follow-up period showing a sufficiently reduced posterior facet.

**Figure 5 ijerph-18-10400-f005:**
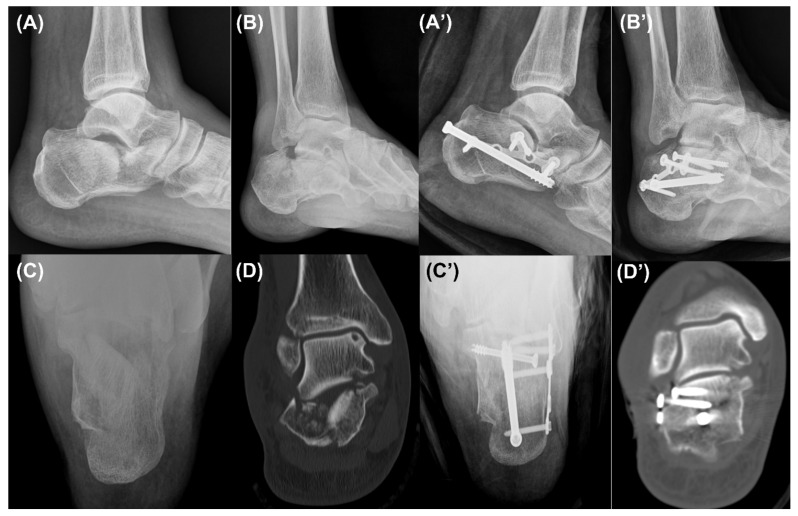
Case 2. Preoperative lateral (**A**), Broden’s (**B**), and axial (**C**) radiographs and a coronal computed tomography image (**D**) showing a Sanders type 4 intra-articular calcaneal fracture. The lateral (**A’**), Broden’s (**B’**), and axial (**C’**) radiographs and a coronal computed tomography image (**D’**) taken in the last follow-up period showing a sufficiently reduced posterior facet.

**Figure 6 ijerph-18-10400-f006:**
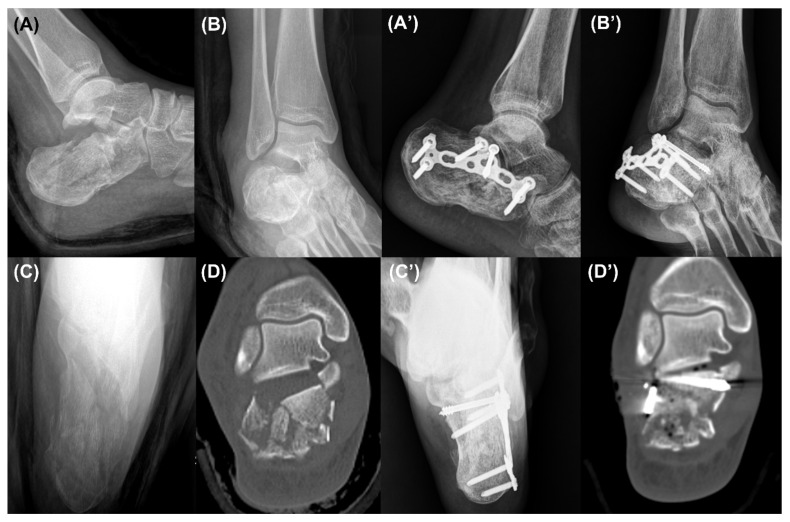
Case 3. Preoperative lateral (**A**), Broden’s (**B**), and axial (**C**) radiographs and a coronal computed tomography image (**D**) showing a Sanders type 4 intra-articular calcaneal fracture. The lateral (**A’**), Broden’s (**B’**), and axial (**C’**) radiographs and a coronal computed tomography image (**D’**) taken in the last follow-up period showing a sufficiently reduced posterior facet.

**Figure 7 ijerph-18-10400-f007:**
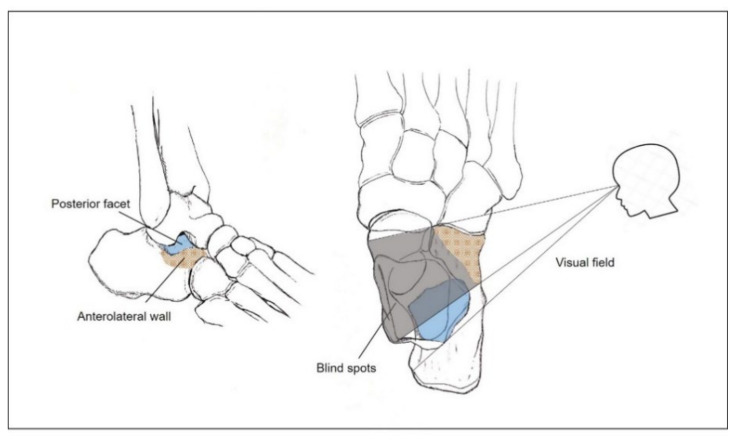
Illustration of the blind spots (gray shadow pattern) created by the anterolateral wall (red colored check pattern) of the calcaneus. Only partial exposure of the posterior facet of the calcaneus (blue colored) becomes possible due to the limited visual field.

**Figure 8 ijerph-18-10400-f008:**
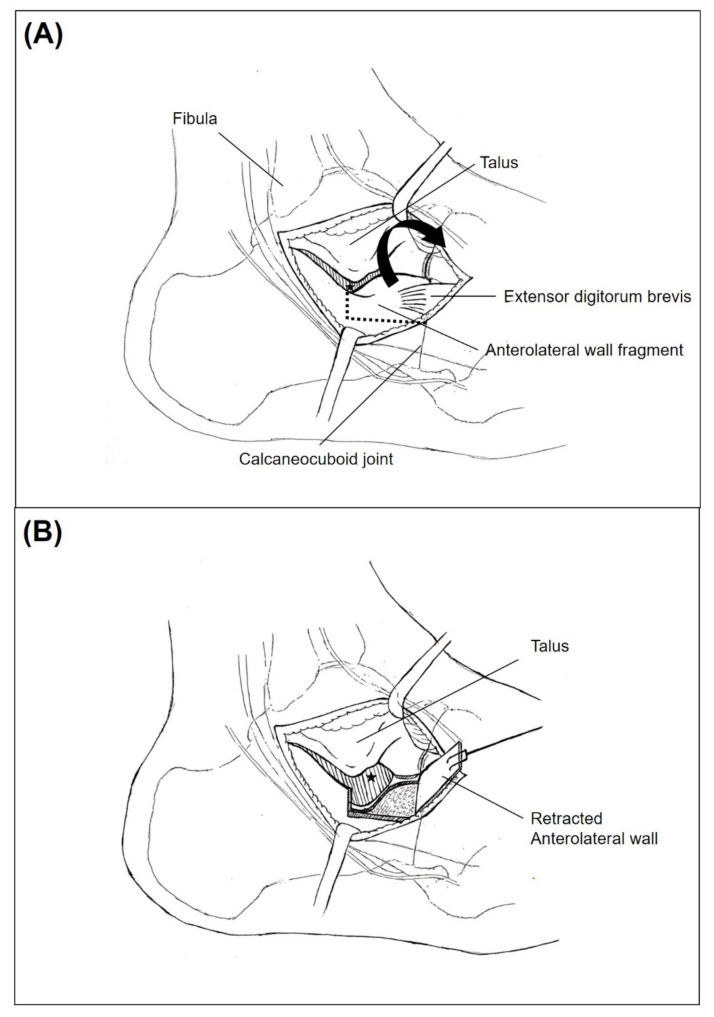
Illustration of the anterolateral fragment open-door technique. (**A**) Anterolateral wall fragment of the calcaneal fracture becomes identified with extensor digitorum brevis muscle attached. Only partial visualization of the posterior facet (Asterisk) with the presence of anterolateral wall. (**B**) After anterosuperior retraction of the anterolateral wall, medial most portion of the posterior facet (Asterisk) becomes visualized.

**Figure 9 ijerph-18-10400-f009:**
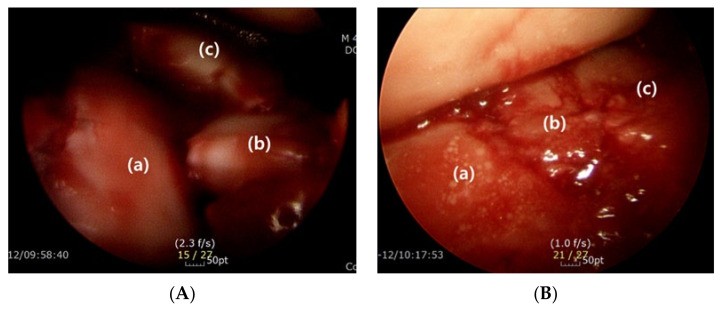
Dry arthroscope image showing articular surface of the posterior facet before (**A**) and after (**B**) the reduction (Lateral (a), middle (b), and innermost (c) fragment of the posterior facet).

**Table 1 ijerph-18-10400-t001:** Demographics.

Characteristics	Frequency Count (%) or Mean ± SD ^1^
Sex	
Male	22 (88%)
Female	3 (12%)
Age (Year)	55.84 ± 11.89
Side	
Right	13 (52%)
Left	12 (48%)
BMI (kg/m^2^)	23.45 ± 2.55
History	
Diabetes	2 (8%)
Smoking	5 (20%)
Injury Mechanism	
Fall from <1 m	4 (16%)
Fall from 1–2 m	14 (56%)
Fall from >2 m	6 (24%)
Traffic accidents	1 (4%)
Sanders Classification	
III	18 (72%)
IV	7 (28%)
Associated other fractures	
Thoracic or Lumbar vertebra	5 (20%)
Patella	1 (4%)
Scaphoid	1 (4%)
Time of injury to surgery (Days)	3.28 ± 1.86

^1^ SD = standard deviation.

**Table 2 ijerph-18-10400-t002:** Radiological results before and after the operation.

	Preoperative	Postoperative	*p*-Value *
Bohler’s angle	5.07 ± 13.43	24.23 ± 7.59	<0.001
Gissane’s angle	109.72 ± 21.10	111.70 ± 6.82	0.6591
Calcaneal height	41.22 ± 5.71	47.31 ± 4.02	<0.001
Calcaneal length	77.51 ± 5.67	78.97 ± 5.73	0.018
Calcaneal width	44.02 ± 4.33	41.38 ± 4.65	0.006
Articular step-off	4.28 ± 2.01	0.76 ± 0.98	<0.001 ^+^
CT Articular step-off	5.65 ± 2.56	0.53 ± 0.83	<0.001 ^+^

* We performed a normality test through the Shapiro–Wilk Test. A parametric test was performed using a paired T-test if there was a normality, and a nonparametric test was performed using Wilcoxon-signed rank test in the case of no normality. ^+^ A Wilcoxon-signed rank test was performed.

**Table 3 ijerph-18-10400-t003:** Postoperative outcome.

Characteristics	Frequency Count (%) or Mean ± SD
Complication	
Superficial infection	2 (8%)
Sural nerve injury	0 (0%)
VAS score	0.92 ± 0.95
AOFAS score	90.08 ± 6.44
Follow up period (Months)	15.28 ± 6.50

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
