# Peer review of "Surgical Treatment Using Sinus Tarsi Approach with Anterolateral Fragment Open-Door Technique in Sanders Type 3 and 4 Displaced Intraarticular Calcaneal Fracture"

_ijerph, 2021, doi:10.3390/ijerph181910400_

Round 1

Reviewer 1 Report

Dear authors, congratulations for this nice work. Some comments about it are the following:

  • Line 92: Please explain how do you create the window (using a saw o just using a chisel, or maybe using the same fracture line??).
  • Figure 2: The quality of the photos are no good enough, I wonder if you can improve it to better understand the technique.
  • Line 133: Please clarify if you check the wounds two days after the surgery because its not the common aftertreatment. On the other hand, why do you use a cast after the surgery if you use a locking plate??
  • Lines 188 and 189: I am very struck by the high average stay of your patients. What do you think is due to? If one of the advantages of the sinus tarsi approach is that we do not have to wait for the soft tissues for the surgery, why is its average stay of 12 days? It should be clarified in the text when discussing its results.
  • Lines 202-203: I am also strongly struck by the excellent result of your series with wound healing, nothing to do with what has been published in the literature and without a doubt surprising . What do you think is due to its good result? Do you carry out any maneuver to protect the tissues different from the rest? Is there any plausible explanation for this?

Many thanks.

Author Response

Dear authors, congratulations for this nice work. Some comments about it are the following:

Line 92: Please explain how do you create the window (using a saw o just using a chisel, or maybe using the same fracture line??).

->Thank you. We clarified the surgical procedure on the manuscript (Line 91-97)

Figure 2: The quality of the photos are no good enough, I wonder if you can improve it to better understand the technique.

-> Thank you for your comment. Although we were able to obtain enhanced visualization of the inner-most portion of the posterior facet, it was handful to take a picture of it even with a dry scope. We improved the quality of the images as best as we could. We hope you would generously understand the difficulties.

Line 133: Please clarify if you check the wounds two days after the surgery because its not the common aftertreatment. On the other hand, why do you use a cast after the surgery if you use a locking plate??

-> Thank you. We revised the postoperative portion of the manuscript (Line 140-148).

Lines 188 and 189: I am very struck by the high average stay of your patients. What do you think is due to? If one of the advantages of the sinus tarsi approach is that we do not have to wait for the soft tissues for the surgery, why is its average stay of 12 days? It should be clarified in the text when discussing its results.

-> Thank you for your comment. As far as we are concerned, we believe that an average of 3.28 days seems to correlate with advantage of sinus tarsi approach. In fact in Korea, the national insurance guarantee lengthy hospital stays of more than a week in terms of calcaneal fracture, which resulted the average stay of 12.36 days. The hospital stay seems be causing confusion, so we find it better to remove the measurement from table 1.

Lines 202-203: I am also strongly struck by the excellent result of your series with wound healing, nothing to do with what has been published in the literature and without a doubt surprising . What do you think is due to its good result? Do you carry out any maneuver to protect the tissues different from the rest? Is there any plausible explanation for this?

-> Thank you for your comment. Most importantly, we speculated that it would be important to preserve the lateral calcaneal artery, which is known to be critical for vascularization of the lateral fasciocutaneous layer. As a result, we extended the incision anteriorly and ceased posteriorly at the level of the fibular tip, which could lower the risk of lateral calcaneal artery damage. We included the details on the discussion portion of the manuscript (Line 314-326)

Reviewer 2 Report

The paper is well written and the figures have a good definition.  The surgical technique is rather new, but 25 cases are few and follow-up is to short.  I suggest you to increase the number of your cases to a minimum of 40 and move the last follow-up to one year minimum.

Other specific considerations:

Line 135: correct "after the surgery" into "after surgery"

Paragraph 2.5. Statistical Analysis: In the methods you include the statistical analysis, but in the results you do not show the statistical analysis.  I believe that a statistical analysis for 25 cases does not have sense.  Please remove this paragraph.

Line 189: You report that the average time of surgery from fracture is 3.28 ± 1.86 days.  After 2 or more days, calcaneal fractures cause serious swelling.  You should specify why you do not operate these patients earlier, and if the swelling does not cause technical difficulties for a mini-invasive technique.

Table 1: The average time of dismission is 12.36 +- 6.1.  You should justify this long hospital stay.

Table 1: Under the table there is a not understandable specification: 1 - Tables may have a footer. Please clarify.

FIGURES:  The illustration of only one case is insufficient.  You should put the figures of 3 cases at least.

Author Response

The paper is well written and the figures have a good definition.  The surgical technique is rather new, but 25 cases are few and follow-up is to short.  I suggest you to increase the number of your cases to a minimum of 40 and move the last follow-up to one year minimum.

-> Thank you and we appreciate your generous comments. We’d improve the weakness of our study in the following study.

Other specific considerations:

Line 135: correct "after the surgery" into "after surgery"

-> Thank you. We revised the manuscript accordingly (Line 143)

Paragraph 2.5. Statistical Analysis: In the methods you include the statistical analysis, but in the results you do not show the statistical analysis.  I believe that a statistical analysis for 25 cases does not have sense.  Please remove this paragraph.

-> Thank you for your comment. We included the statistical analysis because we compared the

preoperative and postoperative calcaneal radiologic measurements using the statistical methods for sample size less than 30 (paired t test or Wilcox-signed rank test).

Line 189: You report that the average time of surgery from fracture is 3.28 ± 1.86 days.  After 2 or more days, calcaneal fractures cause serious swelling.  You should specify why you do not operate these patients earlier, and if the swelling does not cause technical difficulties for a mini-invasive technique.

-> Thank you for your comment. We carried out a surgery as soon as skin condition became stable enough to perform a suture. As a result, the average time of surgery from fracture retrospectively turned out to be 3.28 days. In fact, previous studies have reported that correlation between surgical timing and wound problem is not clear, especially in terms of sinus tarsi approach. We added such explanation in the discussion (Line 322-326).

Table 1: The average time of dismission is 12.36 +- 6.1.  You should justify this long hospital stay.

-> Thank you for your comment. In Korea, the national insurance guarantee lengthy hospital stays of more than a week in terms of calcaneal fracture, which resulted the average stay of 12.36 days. The hospital stay seems be causing confusion, so we find it better to remove the measurement from table 1.

Table 1: Under the table there is a not understandable specification: 1 - Tables may have a footer. Please clarify.

-> Thank you and we revised the manuscript accordingly.

FIGURES:  The illustration of only one case is insufficient.  You should put the figures of 3 cases at least.

-> We appreciate your suggestion and added two more cases on the manuscript.

Reviewer 3 Report

The study is well presented, the intraoperative procedure is well descriped.

  1. The study design is a retrospective analysis. Many ramdomized prospective studies dealing with this topic have been published so far.
  2. The follow-up is varying a lot (12-40 months).
  3. It is not clear whether the anterolateral open door view has been accessed by the accident-caused fracture or iatrogenic by an osteotomy. 
  4. Is would be nice if the chosen postoperativ CT-scans were at the same level than the preoperativ ones. 

Author Response

The study is well presented, the intraoperative procedure is well described.

1. The study design is a retrospective analysis. Many randomized prospective studies dealing with this topic have been published so far.

-> Thank you for your comment. We totally agree with you that many randomized prospective studies of this topic have already been published. However, most of them are focused on less displaced, Sanders type 2 or 3 fractures. In fact, we were able to detect only one randomized study that specifically deals with severely displaced intra-articular calcaneal fractures, that is, Sanders type 3 or 4 fractures (Li et al, Medicine (Baltimore). 2016 Sep; 95(36): e4628.). Even the study includes patients not only with Sanders type 3 or 4 fractures, but also with Sanders type 2. As a result, we’d like to carefully suggest that this study is valuable because the included patients are limited to those with Sanders type 3 or 4 fractures, despite relatively small numbers.

2. The follow-up is varying a lot (12-40 months).

->Thank you for your comment. We included patients who were followed up for at least 12 months with average of 15.28 months. We clarified the manuscript (Line 211-212)

3. It is not clear whether the anterolateral open door view has been accessed by the accident-caused fracture or iatrogenic by an osteotomy. 

-> Thank you. We clarified the surgical procedure on the manuscript (Line 91-97)

4. Is would be nice if the chosen postoperativ CT-scans were at the same level than the preoperativ ones. 

->Thank you for your comment. In fact, owing to the fact that anatomic displacement could be found among the surrounding structures, we tried to reconstruct the images at the similar levels. Additionally, we added two more cases.

Round 2

Reviewer 2 Report

Your revision has improved your manuscript.

Please modify the following parts as indicated:

  • Line 45, together with citation nr 11, add the following citation: Ceccarini P., Manfreda F., Petruccelli R., Talesa G., Rinonapoli G., Caraffa A. “Minimally invasive sinus tarsi approach in Sanders II-III calcaneal fractures in high-demand patients.” Med Glas (Zenica) 18, no 1 (2021): 322-327. doi: 10.17392/1282-21. PMID: 33619940.
  • Line 282. After … (Figure 9), add: differently from the usual arthroscopic procedure, mostly used for the ankle, which necessitate of saline solution (cite Ceccarini P., Rinonapoli G., Antinolfi P., Caraffa A. “Effectiveness of ankle arthroscopic debridement in acute, subacute ankle- bimalleolar, and trimalleolar fractures.” Int Orthop 45 no 3 (2021):721-729. doi: 10.1007/s00264-020-04882-6. Epub 2021. PMID: 33416908; Banerjee S., Gupta A., Elhence A., Choudhary R. “Arthroscopic subtalar arthrodesis as a treatment strategy for subtalar arthritis: A systematic review.” J Foot Ankle Surg 60 no 5 (2021): 1023-1028. doi: 10.1053/j.jfas.2021.04.006. Epub 2021 Apr 20. PMID: 33972158), in this case arthroscopy has to be performed dry, for two reasons: first, the presence of a surgical opening, although if it is a mini-invasive approach, prevents the creation of an internal pressure, second, an increased peri-calcaneal pressure could create damage to the neighboring anatomical structures in a confined space suffering from the hematoma produced by the fracture.
  • Line 211-212 the average follow-up period was 15.28± 6.50, ranging from 12 to ? months. I guess you missed a number. Please correct.
  • Why did you put some citations in bold, some no? Please correct.
  • In the discussion, please add to the already mentioned limits of your study, that the number of the cases should be increased to have a more significant outcome.

Author Response

Line 45, together with citation nr 11, add the following citation: Ceccarini P., Manfreda F., Petruccelli R., Talesa G., Rinonapoli G., Caraffa A. “Minimally invasive sinus tarsi approach in Sanders II-III calcaneal fractures in high-demand patients.” Med Glas (Zenica) 18, no 1 (2021): 322-327. doi: 10.17392/1282-21. PMID: 33619940.

->Thank you and we added the citation you suggested.

Line 282. After … (Figure 9), add: differently from the usual arthroscopic procedure, mostly used for the ankle, which necessitate of saline solution (cite Ceccarini P., Rinonapoli G., Antinolfi P., Caraffa A. “Effectiveness of ankle arthroscopic debridement in acute, subacute ankle- bimalleolar, and trimalleolar fractures.” Int Orthop 45 no 3 (2021):721-729. doi: 10.1007/s00264-020-04882-6. Epub 2021. PMID: 33416908; Banerjee S., Gupta A., Elhence A., Choudhary R. “Arthroscopic subtalar arthrodesis as a treatment strategy for subtalar arthritis: A systematic review.” J Foot Ankle Surg 60 no 5 (2021): 1023-1028. doi: 10.1053/j.jfas.2021.04.006. Epub 2021 Apr 20. PMID: 33972158), in this case arthroscopy has to be performed dry, for two reasons: first, the presence of a surgical opening, although if it is a mini-invasive approach, prevents the creation of an internal pressure, second, an increased peri-calcaneal pressure could create damage to the neighboring anatomical structures in a confined space suffering from the hematoma produced by the fracture.

-> Thank you for your recommendation. We revised the manuscript accordingly (Line 282-288).

Line 211-212 the average follow-up period was 15.28± 6.50, ranging from 12 to ? months. I guess you missed a number. Please correct.

-> Thank you. We revised the manuscript accordingly.

Why did you put some citations in bold, some no? Please correct.

-> Thank you and we revised the manuscript accordingly.

In the discussion, please add to the already mentioned limits of your study, that the number of the cases should be increased to have a more significant outcome.

-> We appreciate your recommendation and revised the manuscript accordingly (Line 350-354).

This manuscript is a resubmission of an earlier submission. The following is a list of the peer review reports and author responses from that submission.

Round 1

Reviewer 1 Report

Thank you for allowing me the read the JCM submission “Surgical treatment of Sanders type 3/4 calcaneal fracture using a sinus tarsi approach with anterolateral fragment ...” this is a retrospective case series study of 25 patients with sander 3/4 DIACF treated with sinus tarsi approach. The paper demonstrated significant correction if Bohler angle, calcaneal width, length, height and articular step using both X-ray and CT scan post operatively. The paper reads well and the low complication rate of 8% is impressive.

 I feel prior to acceptance, but one small issues exist.

  1. Please avoid the use of the use of nonspecific pronouns, such as “there has...” or “ there are...” or “ it is...”

Please see line 181,183,202,281,288,312

  1. Please add the strength of the paper
  2. Please also add future works to end of Discussion.

Author Response

Thank you for allowing me the read the JCM submission “Surgical treatment of Sanders type 3/4 calcaneal fracture using a sinus tarsi approach with anterolateral fragment ...” this is a retrospective case series study of 25 patients with sander 3/4 DIACF treated with sinus tarsi approach. The paper demonstrated significant correction if Bohler angle, calcaneal width, length, height and articular step using both X-ray and CT scan post operatively. The paper reads well and the low complication rate of 8% is impressive.

 I feel prior to acceptance, but one small issues exist.

Please avoid the use of the use of nonspecific pronouns, such as “there has...” or “ there are...” or “ it is...” Please see line 181,183,202,281,288,312

-> Thank you for the comment, We authors agree that there are some inappropriate pronouns presents, and we revised the manuscript thoroughly

Please add the strength of the paper

-> Thank you for the comment, we put additional paragraph on discussion section about the strength of the paper

Please also add future works to end of Discussion.

-> Thank you for the comment. We authors mentioned the future works at the end of Discussion.

“Therefore, succeeding randomized controlled trials or prospective studies to compare the sinus tarsi approach with the extensile lateral approach in the Sanders type 3 or 4 DIACF would be needed to verify our result.”

Reviewer 2 Report

I want to thank you for the opportunity to review this manuscript. After the review and in my humble opinion, the manuscript presents major problems. Below I present my recommendations separated by sections. Hopefully they will be useful:

The English needs to be reviewed by an English speaker.  Some parts were not easy to understand. Several errors are noted with grammar, often using a plural where singular tense is indicated, along with the use of double prepositions, or the omission of a preposition where one is needed

In first section, the title does not match the study described in the manuscript.

The second section, the introduction could be expanded.  It would be helpful to know what the specific problems , and how does this affect function.  In other words, use the published literature and your own thought process to link specific ankleproblems to specific functional problems.

The third section, the method seems adequate.

 Authors must include Ethics consideration section.

The fourth section, Statistical analysis: Why did you do ICC? . You should determine the normality of the sample and after do the statistical test. For this reason it is better to do Mann Withney test Besides authors must include p value in socio demographics characteristics and compares differencies between sex

In addition, the results: The sample is very open, and the results could be conditioned. You are using only visual analysis and don't have tool to measure your result

The discussion section, it seemed as though the discussion was anemic of details as well.  What is the clinical significance?  What is the take-home message you'd like for the reader to gain?  I strongly encourage the authors to review the manuscript for grammatical, spelling and punctuation errors, perfom restructuring of sentences for clarity, add details in the discussion.  

TThe tables and figures not contribute to understanding of the manuscript and not enhance appearance; the information must be rearranged and should be appropriately cited in manuscript.

I think this information is irrelevant and  this manuscript needs to have a lot more information about what was measured, how it was measured, and how it was analyzed before a reader who was not involved in the study (like me) can understand what was done and how the conclusions were drawn.

Author Response

Reviewer 2

I want to thank you for the opportunity to review this manuscript. After the review and in my humble opinion, the manuscript presents major problems. Below I present my recommendations separated by sections. Hopefully they will be useful:

The English needs to be reviewed by an English speaker.  Some parts were not easy to understand. Several errors are noted with grammar, often using a plural where singular tense is indicated, along with the use of double prepositions, or the omission of a preposition where one is needed

-> Thank you for the comment, We authors agree that there are some inappropriate pronouns presents, and we revised the manuscript thoroughly

In first section, the title does not match the study described in the manuscript.

-> Thank you for the comment, we revised the title of our work more clearly.

“Surgical Treatment Using Sinus Tarsi Approach with Anterolateral Fragment Open-Door Technique in the Sanders Type 3 and 4 displaced intraarticular Calcaneal Fracture: A Short-Term Follow-up Study”

The second section, the introduction could be expanded.  It would be helpful to know what the specific problems , and how does this affect function.  In other words, use the published literature and your own thought process to link specific ankle problems to specific functional problems.

->Thank you for the comment, we authors added more specific decriptions about the various benefits and complications of each surgical approach

The third section, the method seems adequate.

-> Thank you for the compliment

 Authors must include Ethics consideration section.

-> Thank you for the advice, we put specific description of our IRB statement

This study was approved by the Institutional Review Board at our institution (IRB number : SCHUH 2020-04-042). and written informed consent for publication of this report was obtained from all the included patients.

The fourth section, Statistical analysis: Why did you do ICC? . You should determine the normality of the sample and after do the statistical test. For this reason it is better to do Mann Withney test Besides authors must include p value in socio demographics characteristics and compares

-> Thank you for the comment. The reason we authors made ICC on our statistical result is mentioned on the manuscript

“All the measurements were performed in the same period by two independent or-thopedics residents, both of whom did not participate in the surgery and were blinded to each other’s findings. Two weeks later, the measurements were repeated in the same fashion in order to determine the intra-observer reliability”

And in case of normality issue, we authors agree that our statistical analysis requires normality test.

So we re-checked our data and made additional normality test, and described on the table footer.

“* We performed normality test through the Shapiro-Wilk Test. Parametric test was performed using paired T-test if there is a normaility, and Non parameteric test was performed using Wil-coxon-signed rank test in case of no normality.

+ Wilcoxon-signed rank test was performed”

However in case of socioeconomic deomographics, since it’s a one sample study comparing pre and post operational outcome, we think our table seems adequate.

In addition, the results: The sample is very open, and the results could be conditioned. You are using only visual analysis and don't have tool to measure your result

-> Thanks for the good pointHowever, methods for measuring the outcome of surgical treatment in fractures have been found in previous literature as well and it mainly consists of clinical (functional) evaluation, radiological evaluation, and evaluation of complications.

Accordingly, in our study, VAS and AOFAS were used as clinical (functional) evaluation items, andBohler’s angle, Gissane’s angle, Calcaneal height, Calcaneal length, Calcaneal width, Articular step-off and CT articular step-off were considered as radiological evaluation items, and various complications were considered.

The discussion section, it seemed as though the discussion was anemic of details as well.  What is the clinical significance?  What is the take-home message you'd like for the reader to gain?  I strongly encourage the authors to review the manuscript for grammatical, spelling and punctuation errors, perfom restructuring of sentences for clarity, add details in the discussion. 

-> Thank you for the comment. Following your advice, we put additional paragraph on Discussion session by amphasizing clinical significance and take home message of our work.

The tables and figures not contribute to understanding of the manuscript and not enhance appearance; the information must be rearranged and should be appropriately cited in manuscript.

-> Thank you for the comment, We comprehensively reviewed the tables and figures, and rechecked the citations in our manuscript

I think this information is irrelevant and  this manuscript needs to have a lot more information about what was measured, how it was measured, and how it was analyzed before a reader who was not involved in the study (like me) can understand what was done and how the conclusions were drawn.

-> We reviewed and revised the entire manuscript for a more detailed and clear conclusion.

Again, We authors are really grateful for your kind advicement on our work. And hope this work can be accepted with all due respect.

Round 2

Reviewer 2 Report

In this reviewed manuscript, researched about   Surgical Treatment Using Sinus Tarsi Approach with Anterol-lateral Fragment Open-Door Technique in the Sanders calcaneal Fracture:

However the research needs to be improved, even though has  an important clinical message, and should  be of  interest to the JCM  readers.  

From my humble point of view this research was well-organized and show precise  results.

However,I recommend review carefully and accurately  this manuscript after major revision because of the following issues.

1.) Title:

The title should say something to impress readers. The  new title is adequate

2.) Introduction

This section shows clear structure with progression on importance of surgical fracture treatment, beside  is a condition frequently described in the literature. Within these sections it was highlighted that prevalence in the population. This led nicely to the purpose of the study.

Them authors  need  discuss this later on in the discussion portion and highlight the findings in the conclusion and abstract.

Introduction may be improved adding new information in order to provide an adequate state-of-the-art including some references. I suggest to include this reference to complete this requirement related to Diabetic foot complications that authors do not included - Marí Serna, et al. "Efficacy of Surgical Techniques in Hallux Abductus Valgus by Application of Scale American Orthopedic Foot and Ankle Society: literature review

https://doi.org/10.17979/ejpod.2018.4.2.3525

3 .) Materials and methods

All methods are  supported according to adequate methodology  how was conducted the study. 

4.) Results

The results section is enough appropriate according to the developed methods and the journal´s scope.

6.) Discussion.

This section needs to be improves in order to understanding of the results section comparing with novel and adequate studies. I would suggest to include information related to foot and ankle surgery Pain for example authors should discuss their result with the achivements of the research of Becerro de Bengoa  et al to complete the trend of research nowadays related to post surgery results

DOI 10.1111/iwj.13400

On the other hand, discussion section include future research studies secondary to the current findings of this study. However authors should discuss their achivement with regard to another surgery parameters such as toes in the case of children

 I suggest to include the following reference to complete this requeriment Bautista Casanova et al DOI 10.3390/jcm9041122

Author Response

In this reviewed manuscript, researched about   Surgical Treatment Using Sinus Tarsi Approach with Anterol-lateral Fragment Open-Door Technique in the Sanders calcaneal Fracture:

However the research needs to be improved, even though has  an important clinical message, and should  be of  interest to the JCM  readers.  

From my humble point of view this research was well-organized and show precise  results.

However,I recommend review carefully and accurately  this manuscript after major revision because of the following issues.

1.) Title:

The title should say something to impress readers. The  new title is adequate

-> Thank you for your response.

2.) Introduction

This section shows clear structure with progression on importance of surgical fracture treatment, beside  is a condition frequently described in the literature. Within these sections it was highlighted that prevalence in the population. This led nicely to the purpose of the study.

Them authors  need  discuss this later on in the discussion portion and highlight the findings in the conclusion and abstract.

Introduction may be improved adding new information in order to provide an adequate state-of-the-art including some references. I suggest to include this reference to complete this requirement related to Diabetic foot complications that authors do not included - Marí Serna, et al. "Efficacy of Surgical Techniques in Hallux Abductus Valgus by Application of Scale American Orthopedic Foot and Ankle Society: literature review

https://doi.org/10.17979/ejpod.2018.4.2.3525

 -> Thank you for the comment. We authors agree on your opinion of the structure of our manuscript, so we put additional statement regarding complications and prevalence including diabetic foot at our Discussion section.

However with all due respect, the previous work which you mentioned is about the Hallux valgus treatment and it lacks information about complication of diabietic foot or wound complication. So please understand that we authors cannot cite this literature.

3 .) Materials and methods

All methods are  supported according to adequate methodology  how was conducted the study. 

  -> Thank you for your response.

4.) Results

The results section is enough appropriate according to the developed methods and the journal´s scope.

-> Thank you for your response.

6.) Discussion.

This section needs to be improves in order to understanding of the results section comparing with novel and adequate studies. I would suggest to include information related to foot and ankle surgery Pain for example authors should discuss their result with the achivements of the research of Becerro de Bengoa  et al to complete the trend of research nowadays related to post surgery results

DOI 10.1111/iwj.13400

-> Thank you for your response. We authors put additional statement about the clinical outcome indicator including previous research in our Discussion section

On the other hand, discussion section include future research studies secondary to the current findings of this study. However authors should discuss their achivement with regard to another surgery parameters such as toes in the case of children

 I suggest to include the following reference to complete this requeriment Bautista Casanova et al DOI 10.3390/jcm9041122

-> Thank you for your response. We authors do agree that previous work of Bautista Casanova et al is well organized, so we referred to the format of this literature about future work.